# Proposal for an Empirical Japanese Diet Score and the Japanese Diet Pyramid

**DOI:** 10.3390/nu11112741

**Published:** 2019-11-12

**Authors:** Masao Kanauchi, Kimiko Kanauchi

**Affiliations:** 1Department of Health and Nutrition, Faculty of Health Science, Kio University, Koryo-cho, Nara 635-0832, Japan; 2Internal Medicine, Nara-Higashi Hospital, Tenri, Nara 632-0001, Japan; benjamin@mahoroba.ne.jp

**Keywords:** dietary quality score, food habits, Japanese diet, Mediterranean diet pyramid

## Abstract

A traditional Japanese diet (JD) has been widely regarded as healthy, contributing to longevity. The modern Japanese lifestyle has become markedly westernized, and it is speculated that the number of people who eat JD is decreasing. A simple evaluation of people with low adherence to JD will help improve dietary life. We developed a simple assessment tool that can capture JD, and examined factors associated with low adherence to JD. A total of 1458 subjects aged 18 to 84 years completed a brief self-administered diet history questionnaire. We constructed an empirical Japanese diet score (eJDS) consisting of 12 items from the common characteristics of a JD. In our participants, 47.7% of subjects reported low adherence to JD and only 11.1% demonstrated high adherence. In multivariate logistic regression analysis, younger age persons, physically inactive persons, and heavy drinkers were associated with low adherence to JD. Based on the cutoff values of eJDS, we proposed to create a Japanese diet pyramid that is easy to use visually. In conclusion, the eJDS and the Japanese diet pyramid will be useful tools for nutrition education and dietary guidance.

## 1. Introduction

The traditional Japanese diet (JD) has been widely regarded as healthy, contributing to longevity and protecting against several noncommunicable diseases (NCD) [1,2]. However, the modern Japanese lifestyle has become markedly westernized, and it is speculated that the number of people who eat traditional JD is decreasing [3]. Several studies have examined factors associated with low adherence to healthy diets, such as Mediterranean diet [4,5,6], because finding a low-adherence group and improving their dietary habits seems to contribute to reducing the risk of NCD. Similarly, the detection of people with low adherence to traditional JD will help improve dietary life, but it has been studied very little in Japan [7]. In addition, the association between JD adherence and nutrient intakes is interesting but has not been considered in detail, so we need to examine further. In order to solve the above problems, a method to detect JD adherence is important. Previously, studies have devised a traditional JD extracted by factor or principal component analysis. Characteristic components include rice [8], miso soup [8,9,10], soybean products [7,8,9,10,11,12,13,14,15], vegetables [7,8,9,10,11,12,13,14,15], fruits [7,12,13,14,15], fish [9,10,11,12,13,15], Japanese pickles [11,12,15], seaweed [9,10,11,12,13,14,15], mushrooms [7,12,14,15], and green tea [7,9,10,11,12,13]. However, approaches using a posteriori dietary pattern analysis include critical methodological issues [16]. The statistical analyses needed in a posteriori dietary pattern analysis are too complicated. Additionally, generalizability to other study populations may be limited, because the dietary patterns defined by factor or principal component analysis are extracted from a selected study population. Therefore, it is important to develop a hypothesis-driven dietary score that can capture a traditional JD and can be applied across different study populations. Although there are some scores to identify a JD [17,18,19,20], they focus on the low-salt diets, overall balance of meals, and main and side dishes. In addition, there is no visual representation of a traditional JD pattern, like the Mediterranean diet pyramid [21,22,23]. The aim of this study is to develop an empirical Japanese diet score (eJDS) as a simple assessment tool, to investigate the relationship between adherence to Japanese diet and nutritional intake, to build a visually comprehensible JD pyramid, and to examine factors associated with low adherence to a traditional JD.

## 2. Methods

### 2.1. Participants

A total of 1607 adults, aged 18 to 84 years, were invited to participate in this study. Participants were recruited from eight workplaces, one local college, and four different areas in central Kinki, Japan. To encourage participation in this study, we took posters or recruitment forms to the college, workplaces, and community-based health classes. Of those invited, 105 persons refused to participate. The remaining 1502 eligible subjects were given a diet history questionnaire. Among them, we excluded 9 subjects who did not complete the questionnaire, 26 subjects who had implausibly low or high estimated caloric intakes (<600 or >4000 kcal per day), and 9 subjects who had missing information for factors needed for statistical adjustment. A total of 1458 participants (781 men, 677 women) were included in this analysis. Participants included 967 working professionals including industrial workers, office workers, formal caregivers, or nursing staff; 233 college students; and 258 community-dwelling adults and elderly (retired, unemployed or housewives).

This study was performed in accordance with the Helsinki Declaration. Study protocols were approved by the Institutional Review Board of Kio University (H26-10), and written informed consent was obtained from each participant.

### 2.2. Dietary Assessment and the Empirical Japanese Diet Score (eJDS)

Using the brief self-administered diet history questionnaire (BDHQ) [24], we examined the frequency of each food intake per week or per day. The intent of the eJDS was to create a dietary score composed of foods frequently found in a traditional JD, building on evidence from previous studies in which specific foods were identified using factor analysis or principal component analysis [7,8,9,10,11,12,13,14,15]. Ten foods identified as components of a traditional JD were rice [8], miso soup [8,9,10], soybean products [7,8,9,10,11,12,13,14,15], vegetables [7,9,10,11,12,13,14,15], fruits [7,12,13,14,15], fish [9,10,11,13,15], Japanese pickles [11,12,15], seaweed [9,10,11,12,13,14,15], mushrooms [7,12,14,15], and green tea [7,8,9,10,11,12,13,14,15]. In addition, we added Japanese-style confections (wagashi) as our 11th component. Although none of the previous studies [7,8,9,10,11,12,13,14,15] reported a low consumption of meat (including meat products) as a characteristic of a JD, it was acknowledged as a traditional dietary habit of before the 1960s [25,26]. We added this as the 12th component of our score. Rice, miso soup, and green tea were evaluated as bowls (cups) per day; soybean products, Japanese pickles, seaweed, mushrooms, fish, meat (including meat products), and Japanese-style confectionery (wagashi) were evaluated by how many times the subjects eat the food per week (by counting as number of meals); and vegetables and fruits were evaluated as servings per day. Each food intake was standardized by z-score and evaluated as follows: z ≥ 0.5, eat often; 0.5 > z ≥ −0.5, moderate; z < −0.5, rarely eat. A cut-off value of eJDS was set with a z score of 0.5 or higher, except for meat and meat products which should be below −0.5. Cut-off values of eJDS were as follows: green tea ≥2 cups/day, rice ≥3 bowls/day, miso-soup ≥2 bowls/day, vegetables ≥5.4 servings/day, fruits ≥1.8 servings/day, fish ≥7 times/week, soy products ≥6 times/week, pickles ≥6 times/week, seaweeds ≥5 times/week, mushroom ≥5 times/week, Japanese confectionery (wagashi) ≥2.5 times/week, and meat and meat products <4 times/week. One point was given to each component if the intake was above the cut-off, except for meat and meat products, where the point was given below the cut-off. The eJDS was calculated as the sum of 12 components, with a possible score of 0 to 12. A higher score indicated better adherence to a traditional JD. The adherence to a traditional JD was classified as follows: low (0–2 components), moderate (3–5 components), and high (≥6 components).

### 2.3. Assessment of Nutrient Intake

Using the BDHQ, values for nutrition were estimated. Combined with standard serving size, the intake frequencies were converted into the average daily intake for each food item. Estimates of nutrients were calculated using an ad hoc computer algorithm for the BDHQ that was based on the corresponding food composition list in the Standard Tables of Food Composition in Japan [27]. Based on this, the associations between adherence to JD and nutrient intake were examined.

### 2.4. The Japanese Diet Pyramid

Using the eJDS cut-off value, a visually easy-to-understand pyramid was constructed with reference to the Mediterranean diet pyramid [21,22,23].

### 2.5. Other Variables

Body mass index (BMI) was calculated as weight in kilograms divided by the square of height in meters. Subjects were classified by BMI category using the Asian standard (BMI <18.5, 18.5–22.9, 23.0–27.4, ≥27.5 kg/m^2^) [28]. A self-reported questionnaire assessed current smoking status (yes, no) and physical activity (active, sedentary). Alcohol consumption was evaluated by BDHQ information and was categorized as none/low (men, <10 g per day; women, <5 g per day), moderate (men, 10–30 g per day; women, 5–15 g per day), and high (men, >30 g per day; women, >15 g per day).

### 2.6. Statistical Analysis

Statistical analysis was performed using SPSS statistics version 21.0 (IBM Corp, Armonk, NY, USA). Continuous variables were described as means ± standard deviation (SD), and the differences between two groups were compared using Student’s *t*-test. Those among three or more groups were compared using one-way analysis of variance. Sample distribution between genders was compared using chi-squared test with standardized residual method. Sample distribution among low, moderate, and high adherence groups were also compared using the chi-squared test. To identify factors associated with low adherence to JD, crude and adjusted (for sex, age classes, BMI levels, smoking status, alcohol intakes, physical activity, and recruited background) odds ratios and 95% confidence intervals were calculated by models of logistic regression. *p*-values < 0.05 were considered statistically significant.

## 3. Results

The eJDS of all subjects ranged from 0 to 10 and no subject had ≥11 score; 11.1% of subjects reported high adherence to JD, whereas 41.2% reported moderate adherence, and 47.7% reported low adherence. Sample distribution of eJDS between genders was not different (*p* = 0.587). Low adherence rates in men and women were 47.9% and 47.6%, respectively. Moderate adherence rates in men and women were 42.8.% and 38.4%, respectively. High adherence rates in men and women were 9.3% and 13.0%, respectively. Distribution of low, moderate, and high adherence between genders were also not different (*p* = 0.069) (Table 1). 

Younger subjects had significantly lower scores than older subjects. Smokers had significantly lower scores than nonsmokers, and physically inactive people had significantly lower scores than active subjects. Moderate- or high-drinkers tended to have low scores. College students had the lowest eJDS, whereas community-dwelling adults and elderly had the highest scores (Table 2). In addition, sample distribution among low, moderate, and high adherence groups were significant in age, smoking habit, physical activity status, alcohol drinking classes and background (Table 2).

A higher adherence to JD was significantly associated with many nutrient intakes and inversely correlated with saturated fat. Although protein intake was correlated with adherence to JD, it was due to fish and soy consumption other than meat. Salt intake increased with the adherence increases. On the other hand, JD adherence was positively correlated with potassium intake (Table 3). 

Based on the cutoff of eJDS, we proposed to create a Japanese diet pyramid (Figure 1). At the base of our pyramid, green tea should be drunk every day. Rice, miso soup, vegetables, and fruits are also core elements to be consumed every day. The weekly inclusion of fish, soy products, pickles, seaweed, mushrooms, and Japanese-style confectionery (wagashi) are recommended, as these foods have many healthful effects. At the top of the pyramid, meat and meat products are foods to limit.

Crude logistic regression analysis revealed the following factors were associated with low adherence to JD: age between 18 and 34 years, those between 35 and 49 years and those between 50 and 64 years compared with those aged ≥65; smokers, as compared with nonsmokers; high alcohol drinkers compared with no/low alcohol drinkers; physically inactive compared with active persons; and college students and workers compared with community-dwelling adults and elderly. Multivariate adjusted logistic regression analysis revealed the following factors were associated with low adherence to JD: age between 18 and 34 years (OR 4.49, 95%CI 2.10–9.61), those between 35 and 49 years (OR 4.15, 95%CI 2.02–8.50) and those between 50 and 64 years (OR 2.36, 95%CI 1.18–4.74) compared with those aged ≥65; physically inactive (OR 1.31, 95%CI 1.05–1.64) compared with active persons; and high alcohol drinkers (OR 1.88, 95%CI 1.40–2.52) compared with no/low alcohol drinkers. In multivariate adjusted analyses, the background and smoking habits were no longer significantly associated with low adherence risk (Table 4).

## 4. Discussion

Our study developed a simple assessment tool (eJDS) that can capture a traditional JD pattern. It draws upon the common components of a traditional JD foods as indicated by literature review [7,8,9,10,11,12,13,14,15]. It should be noted that we incorporated Japanese-style confections (wagashi) into our score. The beneficial effects of wagashi may be attributable to the use of adzuki beans (red beans). Adzuki beans contain relatively high amounts of plant protein, fiber, and polyphenols, and a lower fat content compared with western-style confectionaries. Because a low consumption of meat was a traditional dietary habit before the 1960s in Japan [25,26], we also added this component. Regarding a cutoff of score, many other diet scores (e.g., Mediterranean diet score) have used the median intake of people in their sample [29]. However, it is not always universal, so we used z-score that is a distribution with different standard deviation values replaced by standard normal distribution. We evaluated as follows: z ≥ 0.5, eat often; 0.5 > z ≥ −0.5, moderate; z < −0.5, rarely eat. In general, this may be considered legitimate. We set the cut-off of each food intake with a z-score of 0.5 or higher, except for meat (z < −0.5). Using this score, we examined the associations between adherence to JD and nutrient intake. A higher adherence to JD was significantly associated with many nutrient intakes (protein, carbohydrate, fiber, salt and potassium) and inversely correlated with unsaturated fat. In a study of elderly Japanese [20], the Japanese Diet Index was positively correlated with the intakes of many nutrients (protein, fiber, vitamins A, C, and E, calcium, iron, sodium, potassium, and magnesium) and was negatively correlated with sugar and saturated fat. Another study has also reported that the Japanese Diet Index was positively correlated with the intakes of nutrients (protein, fiber, vitamins A, C, and E, calcium, iron, sodium, potassium, and magnesium) and was negatively correlated with saturated fat [30]. These results suggest that high adherence to JD indicates better overall nutrient intakes, except for sodium. Indeed, the disadvantage of JD is said to be high salt. It is well known that salt intake is positively associated with high blood pressure [31]. Therefore, we recommend using low-salt soy sauce and cooking methods that do not use salt (spices and sourness). On the other hand, potassium intake was highly correlated with JD adherence. Potassium intake from JD may protect against hypertension.

Over the past few decades, the Japanese lifestyle has become markedly more westernized [3]. Unhealthy dietary habits have become more prevalent and switching from a traditional JD to a westernized dietary pattern may play an important role in increases in obesity, metabolic diseases, and cardiovascular disease [8,10,11,12,13,25]. A notable finding of our study was that only 11.1% of subjects reported high adherence to a JD, whereas nearly half of people reported low adherence. Several studies have examined factors associated with low adherence to healthy diets, such as Mediterranean diet [4,5,6]. A study of adults in Morocco reported that obesity, single persons, and persons from rural areas were associated with low adherence to Mediterranean diet [4]. In Spanish women, young age, smoking, and sedentary lifestyle were associated with low adherence to Mediterranean diet [5]. Furthermore, in PREDIMED trials (a primary prevention study conducted in Spain), abdominal obesity, smoking, and single persons were associated with low adherence to Mediterranean diet [6]. On the other hand, to our knowledge, there are no studies using JD scores for factors related to low adherence. However, in a study using the principal component analysis, it has been reported that subjects with lower JD pattern are more likely to be young, men, physically inactive, and current smokers [7]. We evaluated the factors associated with low adherence to JD and reported that younger age persons, physically inactive persons, and high alcohol drinkers are associated with low adherence to the JD. From these perspectives, strategies to encourage a JD should target younger age persons, persons with sedentary behavior, and alcohol drinkers. In a crude model of logistic regression analysis, we also found that smoking is also a factor of low adherence to JD. Combining previous reports with our results, smokers may also be targeted.

Visual and easy-to-understand tools are necessary for dietary guidance and nutrition education. Using the eJDS cut-off value, we created a visual Japanese diet pyramid with reference to the Mediterranean diet pyramid [21,22,23]. Previously, an upside-down Japanese food pyramid (Spinning Top) has been proposed, but it focuses on the overall balance of the meals [18]. Other Japanese diet scores [17,19,20,30] do not suggest graphics like the Mediterranean food pyramid. It is desirable to build a healthy weekly menu by using our pyramid.

This study has some limitations. First to be considered is the setting of a threshold for implausible caloric intake. There are two ways of thinking about underreporting and overreporting calorie intake. One is to calculate EER for each subject and include it in the analysis when the energy intake calculated from BDHQ is 0.5 times or more and less than 1.5 times [32]. Another one is a method that excludes only the intake that is not vague, more specifically, a method based on less than 600 kcal and more than 4000 kcal [33]. It is said that there is a tendency of underreporting in persons with severe obesity, but the Japanese have few severely obese people. We also adopted the latter method. Second, the upper limit of rice consumption must be better defined, as a higher consumption of white rice is reported to correlate to type 2 diabetes [34]. Unfortunately, the upper limit has not been set in our study. Third, this study was conducted among a sample population living in an urban, suburban, and rural setting, but we did not include subjects such as fishermen or residents of mountain villages. Finally, educational history and economic conditions are important factors related to eating habits, but we do not have data on them.

## 5. Conclusions

The empirical Japanese diet score (eJDS) is a simple assessment tool that captures a traditional JD adherence, and higher eJDS positively correlated with good nutrient intake. Only 11.1% of subjects reported high adherence to a JD, whereas nearly half of people reported low adherence. Younger age persons, physically inactive persons, and heavy drinkers were associated with low adherence to JD. Additionally, the Japanese diet pyramid created by the cutoffs of eJDS is a visual and easy-to-understand tool. Further research is needed to demonstrate the effectiveness of dietary guidance and nutrition education using these tools.

## Figures and Tables

**Figure 1 nutrients-11-02741-f001:**
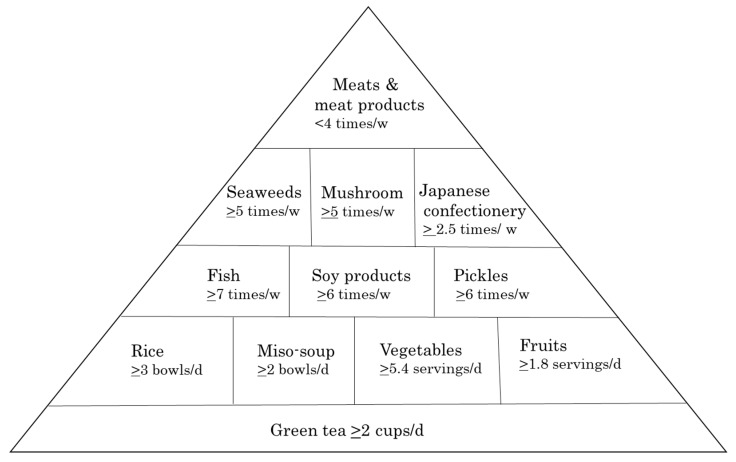
The Japanese diet pyramid.

**Table 1 nutrients-11-02741-t001:** Number (%) of participants meeting each eJDS.

Number of Components	Adherence to JD	Total	Men	Women
0	Low	111 (7.6)	59 (7.6)	52 (7.7)
1		281 (19.3)	143 (18.3)	138 (20.4)
2		304 (20.8)	172 (22.0)	132 (19.5)
3	Moderate	286 (19.6)	158 (20.2)	128 (18.9)
4		184 (12.6)	103 (13.2)	81 (12.0)
5		131 (9.0)	73 (9.3)	58 (8.6)
6	High	84 (5.8)	39 (5.0)	45 (6.6)
7		42 (2.9)	20 (2.6)	22 (3.2)
8		23 (1.6)	10 (1.3)	13 (1.9)
9		10 (0.7)	3 (0.4)	7 (1.0)
10		2 (0.1)	1 (0.1)	1 (0.1)
11		0 (0)	0 (0)	0 (0)
12		0 (0)	0 (0)	0 (0)

eJDS, empirical Japanese diet score; JD, Japanese diet.

**Table 2 nutrients-11-02741-t002:** Factors related to the eJDS.

Factor	Class	*n*	eJDS	*p* ^b^	Low Adherence	Moderate Adherence	High Adherence	*p* ^c^
Sex	Men	781	2.85 ± 1.87	0.323	374 (47.9)	334 (42.8)	73 (9.3)	0.069
	Women	677	2.96 ± 2.05		322 (47.6)	267 (39.4)	88 (13.0)	
Age range (years)	18–34	401	2.45 ± 1.67	0.001	228 (56.9)	151 (37.7)	22 (5.5)	0.001
	35–49	521	2.45 ± 1.76		296 (56.8)	196 (37.6)	29 (5.6)	
	50–64	298	3.12 ± 1.91		126 (42.3)	134 (45.0)	38 (12.8)	
	≥65	238	4.39 ± 2.09		46 (19.3)	120 (50.4)	72 (30.3)	
BMI (kg/m^2^)	<18.5	106	2.94 ± 2.17	0.974	53 (50.0)	40 (37.7)	13 (12.3)	0.710
	18.5–22.9	733	2.91 ± 1.99		356 (48.6)	293 (40.0)	84 (11.5)	
	23.0–27.4	489	2.90 ± 1.92		230 (47.0)	206 (42.1)	53 (10.8)	
	≥27.5	130	2.83 ± 1.74		57 (43.8)	62 (47.7)	11 (8.5)	
Smoker	Yes	287	2.47 ± 1.80	0.001	159 (55.4)	109 (38.0)	19 (6.6)	0.003
	No	1171	3.01 ± 1.98		537 (45.9)	492 (42.0)	142 (12.1)	
Alcohol consumption ^a^	No/Low	961	3.12 ± 2.03	0.001	425 (44.2)	405 (42.1)	131 (13.6)	0.001
	Medium	219	2.72 ± 1.84		103 (47.5)	96 (44.2)	18 (8.3)	
	High	278	2.30 ± 1.62		168 (60.0)	100 (35.7)	12 (4.3)	
Regular exercise	Yes	775	3.11 ± 2.00	0.001	345 (44.5)	325 (41.9)	115 (13.5)	0.003
	No	683	2.67 ± 1.85		351 (51.4)	276 (40.4)	56 (8.2)	
Back-ground	Community-dwelling	258	4.28 ± 2.13	0.001	56 (21.7)	125 (48.4)	77 (29.8)	0.001
	Workers	967	2.63 ± 1.80		507 (52.4)	393 (40.6)	67 (6.9)	
	Students	233	2.49 ± 1.71		133 (57.1)	83 (35.6)	17 (7.3)	

Data are expressed as means ± SD or numbers (%). ^a^ Alcohol consumption was categorized as none or low (men, <10 g per day; women, <5 g per day), medium (men, 10–30 g per day; women, 5–15 g per day), or high (men, >30 g per day; women, >15 g per day). ^b^ By Student’s *t*-test or ANOVA. ^c^ By chi-squared test. BMI, body mass index; eJDS, empirical Japanese diet score.

**Table 3 nutrients-11-02741-t003:** Nutrients by adherence levels to Japanese diet.

Nutrients	Low Adherence	Moderate Adherence	High Adherence	*p*
*n*	696	601	161	
Protein (%E)	14.0 ± 2.5	15.1 ± 2.8	17.4 ± 3.1	0.001
Meat (g/1000 kcal)	41.1 ± 20.2	35.4 ± 18.5	34.1 ± 18.5	0.001
Fish (g/1000 kcal)	35.7 ± 18.2	42.6 ± 24.0	51.4 ± 29.1	0.001
Legume (g/1000 kcal)	21.9 ± 16.4	31.2 ± 21.4	46.9 ± 22.1	0.001
Fat (%E)	27.0 ± 5.9	26.1 ± 5.7	26.3 ± 5.3	0.016
Saturated fat (%E)	7.3 ± 2.1	6.9 ± 1.9	6.8 ± 1.6	0.001
Carbohydrate (%E)	52.6 ± 7.8	54.2 ± 8.1	53.9 ± 7.9	0.002
Dietary fiber (g/1000 kcal)	5.2 ± 1.4	6.5 ± 1.8	8.4 ± 2.0	0.001
Salt (g/1000 kcal)	5.6 ± 1.2	5.9 ± 1.3	6.3 ± 1.3	0.001
Potassium (mg/1000 kcal)	1146 ± 285	1377 ± 376	1731 ± 381	0.001

**Table 4 nutrients-11-02741-t004:** Factors associated with low adherence to JD based on logistic regression analysis.

Factors	Class	Crude OR (95%CI), *p*	Multivariate OR (95%CI), *p*
Sex	Men	1	1
	Women	0.99 (0.80–1.21), 0.902	0.99 (0.76–1.28), 0.925
Age	≥65 years	1	1
	50–64 years	3.06 (2.06–4.54), 0.001	2.36 (1.18–4.74), 0.015
	35–49 years	5.49 (3.81–7.79), 0.001	4.15 (2.02–8.50), 0.001
	18–34 years	5.50 (3.77–8.02), 0.001	4.49 (2.10–9.61), 0.001
BMI	<18.5 kg/m^2^	1	1
	18.5–22.9	0.94 (0.63–1.42), 0.783	0.94 (0.61–1.46), 0.795
	23.0–27.4	0.89 (0.58–1.35), 0.580	0.90 (0.57–1.43), 0.659
	≥ 27.5	0.78 (0.47–1.31), 0.346	0.66 (0.38–1.16), 0.147
Smoking	Nonsmokers	1	1
	Smokers	1.47 (1.13–1.90), 0.004	1.25 (0.94–1.65), 0.128
Alcohol	No/low	1	1
	Moderate	1.14 (0.85–1.53), 0.383	1.18 (0.86–1.61), 0.315
	High	1.89 (1.44–2.48), 0.001	1.88 (1.40–2.52), 0.001
Physical activity	Active	1	1
	Inactive	1.32 (1.07–1.62), 0.009	1.31 (1.05–1.64), 0.018
Background	Community-dwelling	1	1
	Workers	3.97 (2.88–5.48), 0.001	1.37 (0.64–2.95), 0.417
	Students	4.79 (3.24–7.11), 0.001	1.22 (0.63–2.37), 0.548

OR, odds ratio; CI, confidence interval; BMI, body mass index. Adjusted for sex, age classes, BMI levels, smoking status, alcohol intake, physical activity, and background.

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
