# Peer review of "Proposal for an Empirical Japanese Diet Score and the Japanese Diet Pyramid"

_nutrients, 2019, doi:10.3390/nu11112741_

Round 1
Reviewer 1 Report
The aim of this study was to create an empirical Japanese diet score and to examine factors associated with high adherence to a traditional Japanese diet.
Given the general knowledge of nutrition and the knowledge about Japan, this subject is very interesting. Nevertheless, the overall feeling when reading the paper is that three significant questions were too slightly and superficially examined. Although I am convinced of its interest, this paper can be improved.
Introduction.
The introduction focused on the empirical Japanese diet score (eJDS) and I have several issues about it.
What are the clinical issues for a population with a healthy life expectancy of about 70 years (to my knowledge)? What can be the use of this score?
Lines 35-36. Your aim was to examine factors associated with high adherence to a Japanese diet (JD) but:
What are the reasons for studying the associated factors with high adherence to a JD? What is the literature about this? What were your hypotheses?
Part of your discussion is about the food pyramid you proposed but I didn’t read anything about it in the introduction. Wasn’t that also an aim of your paper? I am a little confused because it is mentioned right from the title. I think this part should be introduced:
What already exists? What are the challenges of creating a new one? Why derive the pyramid from the score? What are the challenges to doing so in this way?
Methods.
The “methods” section mainly focused on the factors associated with high adherence and did not sufficiently address the creation of the score and the pyramid. Thus, I have several issues about this section:
Lines 39-41. Is it an exhaustive sample? If so, this should be indicated, otherwise, what is the sample? How did you “choose” the participants to invited? This is an important part to include in the discussion that will follow Lines 39-41 and 46-48. Was the location of recruitment (workplace, college, different areas) used to determine the individual characteristics (workers, students, community-dwelling)? Or was the individual characteristics were determined through the questionnaire? In other words, were there only workers in the workplace location, only students in the college and only community-dwelling in the other areas? If so, multilevel analyses are probably more appropriate. In this way, the location will not be considered as an individual characteristic. Lines 43-44. What are the reasons for setting a caloric intake limit when it is specific to each individual? You considered the <700 and >4500 as implausible but you did not consider the characteristics of the individual. In some cases, 4500 is less implausible than 3000 (e.g. an endurance athlete). I would have rather used the Black method and Goldberg cut-off or other more individual cut-off. In any case, this choice deserves to be discussed in the manuscript.
Lines 57-64. There is something I do not understand about foods included in the eJDS. You started your introduction by “The traditional Japanese diet (JD) has been widely regarded as healthy, contributing to longevity and protecting against several chronic diseases” and I agree with this idea. But I don’t understand why you included Western-style confections while these are not mentioned as part of a traditional JD in the two references mentioned:
Lines 63-64. Why did you include Western-style confections? In terms of health, nuts are highly recommended, sugar-sweetened beverages are to be consumed in a limited quantity… and you did not include these. In terms of traditional JD, are the Western-style confections part of it?
Lines 65-74. The construction of the eJDS is not clear enough for me. To make sure I understood correctly:
Was the score based on the literature mentioned above? Line 70. I understood you considered traditional intakes but what are they? Is there any scientific reason for choosing this 0.33 threshold?
Lines 76-77. Isn’t there a range of IMGs specific to Asian countries? What is the reason for using the international range?
One of the aims of the paper was “to examine factors associated with high adherence to a JD”. For a more reliable examination, I think that confounding factors and effect modifiers should be considered, especially regarding some of your results (see below).
Results.
The result part is clear and easy to read.
Table 1. The p-value of a chi-square would be a plus, I think.
Table 3. When comparing crude and adjusted ORs, some results should be discussed. For example, the difference in category ≥30 of the BMI is significant, from 1.93 (0.81-4.60) to 2.56 (1.01-6.49). The same applied for the age categories. What are the reasons for these differences?
Discussion.
eJDS:
Lines 125-127. Why are the previous tool difficult and why do you need a simple tool? Is it because previous tools ask resources (financial, humans…)? Why do you consider your tool as simple?
Lines 131-133. The construction of eJDS seems you valid but you will need a validation study to discuss about validity. A better description of traditional intakes will also help us to believe in the potential validity. Cutoff values seem reasonable but how did you choose this? How could you determine that it “seems reasonable”? You should develop more this part.
Pyramid:
Lines 133-139. Please refer to the comments below on the pyramid. You should discuss about the strength and weakness of this pyramid in the discussion. Is there any other JD pyramid? Could you make a comparison? It could give strength to your proposal. In addition, what will/could be the use of this pyramid (especially compared to the score)?
Associated factors:
Lines 128-130. The entire results section focuses on the associated factors but there are only 3 lines about it in the discussion part. Your aim was to “examine the associated factors”, therefore, you should further develop this part in the discussion: existing literature, mechanisms behind the association… Right now, you just identified factors.
There is also a lack of discussion about the type of food collection, the sampling or other characteristics not considered that could also be significant factors (e.g.: socioeconomic status).
Conclusion.
Lines 152-154. Your conclusion is only a summary of what you did (without mentioning factors while these could help for interventions…). You should improve this part with key messages, what does all this imply or what needs to be further studied.
Lines 153-154. In addition, you mentioned that it “may help promote healthier eating habits” but you never discussed the association of foods with health.
Author Response
Based on the reviewer comments, there are six major corrections:
1) The number of analysis cases was changed from 1464 to 1458 because the criteria for overestimation and underestimation of calories was revised.
2) Western confectionery was removed from the JD characteristics and Japanese confectionery (Wagashi) was added instead.
3) Since the z-value threshold was reviewed, the cut-off value and pyramid notation for each food were changed.
4) The unit of measurement for vegetables was reanalyzed as a serving (same as for fruits).
5) Since it is more suitable for improving dietary habits and nutrition education to construct a simple score that can extract people with low JD adherence, we changed to calculating low adherence risk.
6) The validation result which sees the relation between JD score and nutrient intake was newly added (new Table 3).
Since the JD score was recalculated and re-analyzed, the results in Table 1-4 were rewritten.
Introduction:
We rewritten the introduction part according to reviewer’s comment. the modern Japanese lifestyle has become markedly westernized, and it is speculated that the number of people who eat traditional JD is decreasing. Since it is more suitable for nutrition improvement and nutrition education to extract people with low adherence to JD, we changed to analyze the factors of low adherence. We also added the purpose of building a pyramid in the introduction part. Although there are some scores to identify a traditional JD, there is no visual representation of a traditional JD patterns like the Mediterranean diet pyramid. This will make it easier to improve eating habits and introduce nutritional guidance.
Methods:
In the revised version, we recalculated with a focus on factors related to low adherence. Our sample is not an exhaustive. To encourage participation in this study, we took posters or recruitment form to their college, workplaces, and community-based health classes. Yes, there were only workers in the workplace location and only students in the college. About the community-dwelling persons, they were retired, unemployed or housewives. The cut-off value for implausible caloric intakes was reset according to research by the BDHQ developers (Prof. Sasaki, Tokyo University). We agreed with the reviewer's opinion, removed Western confectionery from the score and added Japanese confectionery (Wagashi). I decided to use an amended score (including Wagashi) to examine the risk factors for low adherence.
Results:
Yes, I added the p-value of a chi-square in Table 2. Since the score composition was changed and the risk of low adherence was calculated, there was no difference in BMI in the revised version.
Discussion:
Many studies have devised a traditional JD extracted by factor or principal component analysis. The statistical analyses needed in a posteriori dietary pattern analysis are too complicated. Additionally, generalizability to other study populations may be limited, because the dietary patterns defined by factor or principal component analysis are extracted from a selected study population. Therefore, it is better to use a theoretical (a priori) score that is simpler and more like Mediterranean diet score. Certainly, an inverted JD pyramid (top-shaped, Spinning Top) has been proposed, but it focuses on the balance of the whole diet. As far as I know, there is no other JD pyramid. Unfortunately, we do not have data on the subject's economic status or educational history. We added these descriptions to the introduction and discussion part.
Conclusion:
Thank you for your valuable advice. We have added a key message and the need for further research in this part.
Reviewer 2 Report
Introduction:
The introduction is very brief. This could be expanded on to set the scene for the study. In particular more background is needed on why such a tool would be helpful to have and in the Japanese setting.
Methods:
Lines 43-44: The cut-offs for implausible intakes need to be referenced.
Line 53: More information could be given on the diet history, was it interview administrated or self-reported? Did everyone complete one each? Were any multiples done? Was it a validated measurement tool?
Lines 65-69: The various diet components have different measurement units e.g. vegetables calculated as energy-adjusted grams per day where as fruits evaluated as servings per day. Can it be referenced/explained why this is the case? It is stated that a standardised intake was then calculated but I am unsure how these are comparable/standardised if the original measurement units are different, more information on this calculation would also be helpful.
Results:
When talking about "significant" results it would also be helpful to include the number and 95% confidence intervals in brackets in the text.
Discussion:
Overall the discussion needs to reflect and discuss the implications of the results more, at this stage it is very brief. Additionally, if the development of the Japanese Diet Pyramid was an aim of this paper it would be better to state this in your aims/objectives and include in methods and results. The applicability of the pyramid to the public could then be discussed in the discussion. As it is at the moment new ideas and findings are being introduced in the discussion which should not be the case.
Line 132-133 "Cutoff values, determined by a standardized z value, also seem reasonable" needs justification.
Line 138 "This pyramid seems easily introducible into clinical practice." this comment needs further discussion, how and why is it easy to introduce into clinical practice, what are the likely implications?
Author Response
Based on the reviewer comments, there are six major corrections:
1) The number of analysis cases was changed from 1464 to 1458 because the criteria for overestimation and underestimation of calories was revised.
2) Western confectionery was removed from the JD characteristics and Japanese confectionery (Wagashi) was added instead.
3) Since the z-value threshold was reviewed, the cut-off value and pyramid notation for each food were changed.
4) The unit of measurement for vegetables was reanalyzed as a serving (same as for fruits).
5) Since it is more suitable for improving dietary habits and nutrition education to construct a simple score that can extract people with low JD adherence, we changed to calculating low adherence risk.
6) The validation result which sees the relation between JD score and nutrient intake was newly added (new Table 3).
Since the JD score was recalculated and re-analyzed, the results in Table 1-4 were rewritten.
Introduction:
Thank you for your valuable advice. We have added some background on the useful of our score and pyramid in this part.
Methods:
The cut-off value for implausible caloric intakes was reset according to research by the BDHQ developers (Prof. Sasaki, Tokyo University). The BDHQ survey is self-filled and no interviews are conducted. The BDHQ is a validated method used in many studies. We agree with the reviewer's opinion and the unit of measurement for vegetables was reanalyzed as a serving (same as for fruits).
Results:
Yes, we added the OR and 95%CI in brackets in the text.
Discussion:We agree with the reviewer's opinion. We added to the purpose and method session about JD pyramid development. We reviewed the z-value threshold in the revised version and recalculated the JD score based on the new cut-off value for each food. We have also changed the notation of the pyramid. We have described in more detail the usefulness and application of pyramids.
Reviewer 3 Report
A very interested work for creating an empirical Japanese Diet score. It is a straight-forward paper. I would suggest that the authors explain a bit further their findings in the discussion section and maybe compare and contrast it further with relative literature.
For example, the authors state that a notable finding of this study was that the adherence rate to JD is relatively low in modern Japan and that older age, obesity, nonsmokers, and physically active persons are associated with high adherence to the JD. How do they explain the above?
I think this will strengthen the manuscript.
Author Response
Based on the reviewer comments, there are six major corrections:
1) The number of analysis cases was changed from 1464 to 1458 because the criteria for overestimation and underestimation of calories was revised.
2) Western confectionery was removed from the JD characteristics and Japanese confectionery (Wagashi) was added instead.
3) Since the z-value threshold was reviewed, the cut-off value and pyramid notation for each food were changed.
4) The unit of measurement for vegetables was reanalyzed as a serving (same as for fruits).
5) Since it is more suitable for improving dietary habits and nutrition education to construct a simple score that can extract people with low JD adherence, we changed to calculating low adherence risk.
6) The validation result which sees the relation between JD score and nutrient intake was newly added (new Table 3).
Since the JD score was recalculated and re-analyzed, the results in Table 1-4 were rewritten.
Thank you for your valuable advice. We explained the discussion in more detail and cited the relevant literature further. In the revised version, we changed to calculating low adherence risk, because it is more important to find people with low adherence and clarify their risk factors. As a result of the reanalysis, younger, less physically active and heavy drinking were extracted as factors related to low adherence risk (additionally, smoking in crude model).
Round 2
Reviewer 1 Report
The manuscript has been improved but there are still many questions (mentioned in the previous round) unanswered. The manuscript is straight to the point but some aspects are not developed enough to understand the value of the work.The manuscript would be much better if all the objectives were sufficiently introduced, justified and discussed.
Since there were no specific answers to my questions, it is not impossible that I may ask a question to which there has been an amendment.
Introduction.
I still don't understand the value of studying the factors associated with low adherence.
What is the literature about this? What were your hypotheses?
Methods.
The "methods" section has been greatly improved but there are still some unanswered questions.
According to your answer regarding recruitment, why didn't you do a multilevel analysis? Don't you think that there may be a "recruitment effect"? It seems more correct to me than just adjusting for "background".
My question about the threshold of caloric intake was more about the fact that it is "set at" than the values used. I am not convinced that using set values (whatever they may be) is the best choice. I remain convinced that these values must be personal as caloric intake is very individual. At least this should be discussed in the "discussion" section. Is there any research about the threshold used?
Does the amended score corresponded to the 0.44 cutoff value? Again, is there any scientific reason for choosing this 0.44 threshold?
Examining the association between adherence to JD and nutrient intake is very interesting. But this should be more introduced and discussed. A little more methodology on this would also be a plus. Isn't it a new aim of this paper?
Again, isn’t there a range of IMGs specific to Asian countries? What is the reason for using the international range?
Results.
Table 1. As previously said, the p-value of a chi-square would be a plus, I think.
Discussion.
Line 164. I am not sure about the term "undesirable nutrient".
Have there ever been any studies on the association between nutrients and JD? If so, it could strengthen the results, if not, it is interesting to note this.
Lines 169-176. Some ideas should be introduced (in the introduction section). This would help to better understand the value of this work. In addition, it should be better developed. Again, your aim was to “examine the associated factors”, therefore, you should further develop this part in the discussion: existing literature, mechanisms behind the association… Right now, you just identified factors.
Again, there is also a lack of discussion about the type of food collection, the sampling, the caloric intake cutoff, the 0.44 cutoff, ...
Author Response
Thank you for your valuable advice. We rewrote our manuscript based on the reviewer’s opinion, except for the additional statistical analysis. I am sorry that we could not perform the statistical analysis. I also think that the recruitment background is a confounding factor. We performed the crude and multivariate adjusted logistic regression analysis. Certainly, in crude analysis there was a link between the recruited background and low adherence risk. In multivariate adjusted model, the recruited background was no longer significantly associated with low adherence risk. Other points pointed out have been rewritten as follows:
Introduction
We have added the meaning and literature to study the factors related to low adherence: Several studies have examined factors associated with low adherence to healthy diets, such as Mediterranean diet [4-6], because finding a low-adherence group and improving their dietary habits seems to contribute to reducing the risk of NCD. Similarly, detecting of people with low adherence to traditional JD will help improve dietary life.
El Rhazi K, Nejjari C, Romaguera D, Feart C, Obtel M, Zidouh A, Bekkali R, Gateau PB. Adherence to a Mediterranean diet in Morocco and its correlates: cross-sectional analysis of a sample of the adult Moroccan population. BMC Public Health 2012, 12, 345. doi: 10.1186/1471-2458-12-345. Olmedo-Requena R, Fernández JG, Prieto CA, Moreno JM, Bueno-Cavanillas A, Jiménez-Moleón JJ. Factors associated with a low adherence to a Mediterranean diet pattern in healthy Spanish women before pregnancy. Public Health Nutrition, 2014, 17, 648-56. doi: 10.1017/S1368980013000657. Hu EA, Toledo E, Diez-Espino J, Estruch R, Corella D, Salas-Salvado J, Vinyoles E, Gomez-Gracia E, Aros F, Fiol M, Lapetra J, Serra-Majem L, Pintó X, Portillo MP, Lamuela-Raventos RM, Ros E, Sorli JV, Martinez-Gonzalez MA. Lifestyles and risk factors associated with adherence to the Mediterranean diet: a baseline assessment of the PREDIMED trial. PLoS ONE, 2013, 8, e60166. doi: 10.1371/journal.pone.0060166.
Methods (+Discussion)
I am sorry that we could not perform the statistical analysis, while I think that the recruited background is a confounding factor. Other points pointed out have been rewritten.
I agree your comment and rewrote about the threshold of caloric intake: This study has some limitations. First, there are two ways of thinking about underreporting and overreporting calorie intake. One is to calculate EER for each subject and include it in the analysis when the energy intake calculated from BDHQ is 0.5 times or more and less than 1.5 times [32]. Another one is a method that excludes only the intake that is not vague, more specifically, a method based on less than 600 kcal and more than 4000 kcal [33]. It is said that there is a tendency of underreporting in persons with severe obesity, but the Japanese have few severely obese people. We also adopted the latter method.
Sasaki S, Katagiri A, Tsuji T, Shimoda T, Amano K. Self-reported rate of eating correlates with body mass index in 18-y-old Japanese women. Int J Obes Relat Metab Disord 2003, 27, 1405-10. Murakami K, Sasaki S, Takahashi Y, Okubo H, Hosoi Y, Horiguchi H. Oguma E, Kayama F. Dietary glycemic index and load in relation to metabolic risk factors in Japanese female farmers with traditional dietary habits. Am J Clin Nutr 2006, 83, 1161-9.
With the reviewer's opinion, we changed the z-score criterion and rewritten new data for it. We added it to the discussion: Regarding a cutoff of score, many other diet score (e.g. Mediterranean diet score) have used the median intake of people in their sample [29]. But it's not always universal, so we used z-score that is a distribution with different standard deviation values replaced with standard normal distribution. We evaluated as follows: z-score ≥0.5, eat often; 0.5>z≥-0.5, moderate; z<-0.5, rarely eat. In general, it may be considered legitimate.
We have described in more detail the relationship between JD adherence and nutrient intake in the method and discussion:
2.3. Assessment of nutrient intake
Using the BDHQ, values for nutrition were estimated. Combined with standard serving size, the intake frequencies were converted into the average daily intake for each food item. Estimates of nutrients were calculated using an adhoc computer algorithm for the BDHQ that based on the corresponding food composition list in the Standard Tables of Food Composition in Japan [27]. Based on it, the associations between adherence to JD and nutrient intake were examined.
In a study of elderly Japanese, the Japanese Diet Index was positively correlated with the intakes of many nutrients (protein, fiber, vitamin A, C and E, calcium, iron, sodium, potassium and magnesium) and was negatively correlated with sugar and saturated fat [20]. Another study has also reported that the Japanese Diet Index was positively correlated with the intakes of nutrients (protein, fiber, vitamin A, C and E, calcium, iron, sodium, potassium and magnesium) and was negatively correlated with saturated fat [30]. These results suggest that high adherence to JD indicate better overall nutrient intakes, except for sodium.
We used the BMI criteria for Asians:
Subjects were classified by BMI category using the Asian standard (BMI < 18.5, 18.5–22.9, 23.0–27.4, ≥ 27.5 kg/m2) [28].
28. WHO Expert Consultation. Appropriate body-mass index for Asian populations and its implications for policy and intervention strategies. Lancet 2004, 363(9403):157-63. doi: 1016/S0140-6736(03)15268-3
Results
Yes, we added the data (p-value by chi-square test) into the results and Table 2:
In addition, sample distribution among low, moderate and high adherence groups were significant in age, smoking habit, physical activity status, alcohol drinking classes and background (Table 2).
Discussion
Yes, we deleted the term “undesirable”.
Yes, there are reports on the relationship between nutrients and JD, so I discussed in detail:
In a study of elderly Japanese, the Japanese Diet Index was positively correlated with the intakes of many nutrients (protein, fiber, vitamin A, C and E, calcium, iron, sodium, potassium and magnesium) and was negatively correlated with sugar and saturated fat [20]. Another study has also reported that the Japanese Diet Index was positively correlated with the intakes of nutrients (protein, fiber, vitamin A, C and E, calcium, iron, sodium, potassium and magnesium) and was negatively correlated with saturated fat [30]. These results suggest that high adherence to JD indicate better overall nutrient intakes, except for sodium.
Yes, we added in the introduction about investigating factors related to low adherence:
Several studies have examined factors associated with low adherence to healthy diets, such as Mediterranean diet [4-6], because finding a low-adherence group and improving their dietary habits seems to contribute to reducing the risk of NCD. Similarly, detecting of people with low adherence to traditional JD will help improve dietary life.
Since the JD score was recalculated and re-analyzed, the results in Table 1-4 and Figure 1 were rewritten. The numbers have changed slightly, but the conclusions have not changed significantly.
Reviewer 2 Report
Thank you for considering my comments, your revised manuscript is much improved.
I think it would be worthwhile to add more discussion on the increased salt intake with increased adherence to JD. In terms of NCD risk there is high quality evidence that shows the link between increasing salt intake and increasing blood pressure. Therefore, the risk of suggesting this diet should be detailed more specifically in lines 162-167 of the manuscript, and perhaps recommendations should be more conservative, and thought could be given on how salt intake could be reduced while still adhering to JD (e.g. use of salt substitutes or low salt alternatives).
Author Response
Thank you for your valuable advice. We rewrote our manuscript according to the reviewer’s opinion.
I agree your comment. I described salt intake and blood pressure risks, and added recommendations on how to reduce salt:
These results suggest that high adherence to JD indicate better overall nutrient intakes, except for sodium. Indeed, the disadvantage of JD is said to be high salt. It is well known that salt intake is positively associated with high blood pressure [31]. Therefore, we recommend using low-salt soy sauce and cooking methods that do not use salt (spices and sourness). On the other hand, potassium intake was highly correlated with JD adherence. Potassium intake from Japanese diet may protect against hypertension.
(Based on another reviewer’s comment, the JD score was recalculated and re-analyzed. The results in Table 1-4 and Figure 1 were rewritten. The numbers have changed slightly, but the conclusions have not changed significantly.)
Round 3
Reviewer 1 Report
The manuscript has really been improved and the answers to my questions are much clearer, thank you. However, I still have a few small points to improve.
Introduction
We have added the meaning and literature to study the factors related to low adherence: Several studies have examined factors associated with low adherence to healthy diets, such as Mediterranean diet [4-6], because finding a low-adherence group and improving their dietary habits seems to contribute to reducing the risk of NCD. Similarly, detecting of people with low adherence to traditional JD will help improve dietary life.
Thank you.
I think that each objective should be discussed in each part of the manuscript (introduction, methodology, results, discussion). As previously said, the association between adherence to JD and nutrient intake should be introduced. It is difficult to understand why this is being studied if it is not first introduced.
Methods (+Discussion)
I am sorry that we could not perform the statistical analysis, while I think that the recruited background is a confounding factor. Other points pointed out have been rewritten
I still think that multilevel analyses would be more appropriate and that you can perform these analyses, but considering the background in multivariate analysis is better than nothing. That's ok
I agree your comment and rewrote about the threshold of caloric intake: This study has some limitations. First, there are two ways of thinking about underreporting and overreporting calorie intake. One is to calculate EER for each subject and include it in the analysis when the energy intake calculated from BDHQ is 0.5 times or more and less than 1.5 times [32]. Another one is a method that excludes only the intake that is not vague, more specifically, a method based on less than 600 kcal and more than 4000 kcal [33]. It is said that there is a tendency of underreporting in persons with severe obesity, but the Japanese have few severely obese people. We also adopted the latter method.
Thank you for the explanations.
With the reviewer's opinion, we changed the z-score criterion and rewritten new data for it. We added it to the discussion: Regarding a cutoff of score, many other diet score (e.g. Mediterranean diet score) have used the median intake of people in their sample [29]. But it's not always universal, so we used z-score that is a distribution with different standard deviation values replaced with standard normal distribution. We evaluated as follows: z-score ≥0.5, eat often; 0.5>z≥-0.5, moderate; z<-0.5, rarely eat. In general, it may be considered legitimate.
There have been 3 different cut-off versions since the beginning (0.33, 0.44 and 0.5) while I was not asking to change it but to justify it. This makes me confused about this score and the potential validity. At least, the last threshold has been justified.
Results
Yes, we added the data (p-value by chi-square test) into the results and Table 2:
Since the first round, I am writing about Table 1, not Table 2. I think it is interesting to know how men are doing compared to women. This can only be a * to mention where there are significant differences.
Discussion
Several studies have examined factors associated with low adherence to healthy diets, such as Mediterranean diet [4-6], because finding a low-adherence group and improving their dietary habits seems to contribute to reducing the risk of NCD. Similarly, detecting of people with low adherence to traditional JD will help improve dietary life.
Thank you for the improvement. So there is no literature about this in Japan?
Author Response
Thank you for your valuable advice. We rewrote our manuscript based on your opinion.
Introduction
Yes, we added the meaning of studying the association between JD adherence and nutrient intakes:
“Similarly, detecting of people with low adherence to traditional JD will help improve dietary life, but it has been studied very little in Japan [7]. In addition, the association between JD adherence and nutrient intakes is interesting. But it is not considered in detail, so we need to examine further. In order to solve the above problems, a method to detect JD adherence is important.”
As a result of changing the text description, reference #12 was replaced with number 7, so the following reference numbers were changed (#8 to #12 in revised version).
Methods
Thank you for your permission about the statistical analysis and threshold of cut-off.
“We set the cut-off of each food intake with a z-score of 0.5 or higher, except for meat (z<-0.5).”
In the method section, the cut-off value for each food was not listed, so I think it is difficult to understand (Although it is filled in Figure 1, it is difficult to understand without seeing the figure). We added the cut-off value of each food to the text in this revised manuscript.
Results (Table 1)
Yes, we described in more detail the description for the difference between men and women. We reanalyzed using chi-square with residual analysis. In the results, each score of eJDS and each grade of JD adherence did not differ between men and women. In addition, since the correspondence between the text in the result part and Table 1 was insufficient and difficult to understand, the composition of Table 1 was slightly arranged.
“Sample distribution between genders was compared using chi-square test with standardized residual method. Sample distribution among low, moderate and high adherence groups were also compared using the chi-square test. “(into the part of statistical analysis)
“Sample distribution of eJDS between genders was not different (p=0.587). Low adherence rate in men and women were 47.9% and 47.6%, respectively. Moderate adherence rate in men and women were 42.8.% and 38.4%, respectively. High adherence rate in men and women were 9.3% and 13.0%, respectively. Distribution of low, moderate and high adherence between genders were also not different (p=0.069) (Table 1).” (into the part of results)
Discussion
To our knowledge, there are no studies using JD scores for factors related to low adherence. However, in a study using the principal component analysis, it has been reported that subjects with lower JD pattern are more likely to be young, men, physically inactive, and current smokers [7]. We added the description to the introduction part and the discussion part:
“On the other hand, to our knowledge, there are no studies using JD scores for factors related to low adherence. However, in a study using the principal component analysis, it has been reported that subjects with lower JD pattern are more likely to be young, men, physically inactive, and current smokers [7]. We evaluated the factors associated with low adherence to JD and reported that younger age, physically inactive persons and high alcohol drinkers are associated with low adherence to the JD. From these perspectives, strategies to encourage a JD should target younger age, persons with sedentary behavior and alcohol drinkers. Although in a crude model of logistic regression analysis, we also found that smokers are a factor of low adherence to JD. Combining previous reports with our results, smokers may also be the target.”